# A Perspective on the Quality of Life of Hemophilia A Patients in Romania—A Study on 100 Patients

Catalina Guran [1,2,3,*], Hortensia Mărioara Ioniţă [1,2,3], Mieta-Gabriela Haţegan [4], Ioana Ioniţă [1,2,3], Adina Trăilă [5] and Alina-Maria Ilie [3]

1 Department of Hematology, University of Medicine and Pharmacy Victor Babes Timisoara, Piata Eftimie Murgu nr 2, 300041 Timisoara, Timis, Romania
2 Centrul de Cercetare Multidisciplinară a Hemopatiilor Maligne, University of Medicine and Pharmacy Victor Babes Timisoara, Piata Eftimie Murgu nr. 2, 300041 Timisoara, Timis, Romania
3 Hematology Clinic, Municipal Emergency Hospital Timisoara, Str. Gheorghe Dima nr. 5, 300079 Timisoara, Timis, Romania
4 Psychiatry Hospital Gataia, 307185 Gataia, Timis, Romania
5 Medical Center for Children and Youngsters "Cristian Serban" Buzias, Str. Avram Iancu, nr 18, 305100 Buzias, Timis, Romania
* Correspondence: hategan.catalina@umft.ro

**Abstract:** Hemophilia A is an X-linked coagulopathy, where there is a deficit in the production of the coagulation factor VIII. Even though there is a higher incidence of hemophilia A than of hemophilia B, it is still considered a rare disease, as its incidence is of 1 in 10,000 people born. We have applied three questionnaires regarding quality of life: Haem-A-QoL, Haemo-SYM and EQ-5D-5L to 101 adult patients with hemophilia A, which were separated into two groups: on-demand and prophylactic treatment. The results showed a relatively young lot, with medium and high education, but with a sedentary lifestyle and are pensioners. They also seem to have moderate mobility issues but, overall, a good quality of life. The quality of life in our studied lot is relatively good, but a more diverse lot is advised.

**Keywords:** quality of life; QoL; hemophilia; hemophilia A; Haem-A-QoL; EQ-5D-5L; Haemo-SYM

## 1. Introduction

Hemophilia A can be classified as mild hemophilia when serum levels of factor VIII are over 5%, medium between 1 and 5% and severe when serum levels are under 1% [1]. In Romania, prophylactic treatment was approved and funded in 2017 [2], with patients enrolled at the beginning of 2018 due to legislative issues. Therefore, measuring quality of life was not the physicians' primary concern at that moment in time. There have been some papers written on the quality of life of children with hemophilia A, which showed a low quality of life in children and teens in our country, with most of them already having chronic joint involvement in the age range of 6–12 years [3]. To the best of our knowledge, no similar studies have been performed on adult patients; therefore, our study is the first to try and assess the results obtained by prophylactic treatment programs. These results can help us understand if the program is effective and what improvements can be made to it.

## 2. Materials and Methods

The study was conducted in three Romanian medical centers, one of which is a center of excellence for hemophilia treatment. The data were collected over a period of 6 months, from November 2021 to April 2022. The inclusion criteria were adult males (over 18 years of age), with a hemophilia A diagnosis, who were admitted to the medical centers at the time of our study. Excluded from the study were patients who could not read or write, and therefore unable to complete the questionnaires; those under the age of 18; and those that

did not agree to participate in the study. We gave every patient a consent form and 3 self-complete instruments: Haem-A-QoL, Haemo-SYM and EQ-5D-5L. Moreover, every patient was interviewed and socio-demographic data were collected from them, and from their medical records. All this information was registered in an Excel sheet and was analyzed with the statistical program Epi-Info7.2.5.0. (Center of Disease Control, Atlanta, GA, USA).

The functional deficit was defined by the Handicap Committee, according to the definitions given by the Ministry of Labor in 2007. These were developed in accordance with World Health ORganization (WHO) criteria from 2007. These were defined as no deficit, mild, moderate, severe and grave. The mild deficit is defined as not having any physical limitations, or having mild limitations, being able to undergo daily chores and work activities, which corresponds to a full-time schedule. The moderate functional deficit is defined as physical activity being moderately limited, patients are able to undergo daily activities but only in certain conditions, which corresponds with a half-time work schedule. Severe functional deficit is defined by moderate or severe limitations of activities. The patient can perform mild or sedentary activities. Usually, professional life is excluded, but the capacity for self-serve is preserved. The grave functional deficit is defined by the patient being unable to perform any activities and being completely dependent on a caregiver. The data regarding functional deficit were collected from patient charts, as per the decisions given by the Handicap Committee [4].

The calculation of the score in Haim-A-QoL is performed by transforming the scores achieved in each dimension on scales ranging from zero to 100, with 0 representing the best Quality of life (QoL), and 100 representing the worst QoL. The total score utilized was the standardized scale score, which was obtained by adding all the transformed scores in each domain [5], which results in a range of scores between 0 and 1000, with 0 being the best QoL and 1000 being the worst QoL.

For the EQ-5D-5L, we utilized the manual provided by the EuroQoL foundation [6]. The self-evaluation scale is between 0 and 100, with 0 being the worst QoL and 100 being the best QoL. Each dimension in the questionnaire has five response levels: no problems, slight problems, moderate problems, severe problems and unable to/extreme problems. Responses are coded as single-digit numbers expressing the severity level selected in each dimension. For instance, 'slight problems' (e.g., 'I have slight problems in walking about') is always coded as '2'. The digits for the five dimensions can be combined in a 5-digit code that describes the respondent's health state; for instance, 21111 means slight problems in the mobility dimension and no problems in any of the other dimensions. These numbers were transformed into index values, which are derived by applying a formula that attaches values (weights) to each of the levels in each dimension. The index is calculated by deducting the appropriate weights from 1, the value for full health (i.e., state 11111).

For the Haemo-Sym, the values are obtained by adding the corresponding numbers to each answer given by the patient. The scores obtained are between 0 and 95, with 0 being the best QoL and 95 being the worst QoL.

## 3. Results

Our lot consisted of 101 male, adult patients, aged between 20 and 70 years, with a mean of 41 years of age, of which 51 patients were without and 50 were with prophylactic treatment. The description of our lot is shown in Table 1 below.

A total of 80.20%, meaning 81 patients, had severe hemophilia A, data which are similar to those found in the international literature [7–9]; 17.82% had moderate hemophilia, and only 1.98%, 2 patients, were mild cases.

More than half of the patients in our lot presented viral infections, as seen in Table 2 below. There were 59 patients with Hepatitic Virus C (HVC), 3 with Hepatitic Virus B (HVB) and 2 with both viral infections. Only 36 out of 101 patients had no viral infections.

**Table 1.** Lot description.

|  |  | Median | Standard Deviation | Min | Max |
|---|---|---|---|---|---|
| Age |  | 41.17 | 11.05 | 20 | 70 |
| Disease onset |  | 1.91 | 1.99 | 0 | 14 |
| Years of evolution |  | 39.26 | 10.90 | 18 | 66 |
| Hemorrhagic events |  | 46.63 | 27.49 | 5 | 150 |
| Home | Rural | 56.44 | 46.2 | 66.28 |  |
|  | Urban | 43.56 | 33.72 | 53.8 |  |
| Studies | Inferior levels | 36.53 | 27.27 | 46.81 |  |
|  | Medium and superior levels | 63.37 | 53.19 | 72.73 |  |
| Occupation | Employed | 31.68 | 22.78 | 41.69 |  |
|  | Non-employed | 68.32 | 58.31 | 77.22 |  |
| Marital status | Single | 35.64 | 26.36 | 45.79 |  |
|  | Married | 53.47 | 43.27 | 63.45 |  |
|  | Divorced/widowed | 10.98 | 5.56 | 18.65 |  |
| Form of disease | Severe | 80.2 | 71.09 | 87.46 |  |
|  | Moderate | 17.82 | 10.92 | 26.7 |  |
|  | Mild | 1.98 | 0.24 | 6.97 |  |
| Way of life | Sedentary | 68.32 | 58.31 | 77.22 |  |
|  | Active | 31.68 | 22.78 | 41.69 |  |

**Table 2.** Viral infection distribution in our lot.

| INF | Frequency | Percent | Cum. Percent |
|---|---|---|---|
| HVB | 3 | 2.97% | 2.97% |
| HVC | 59 | 58.42% | 61.39% |
| HVC, HVB | 3 | 2.97% | 64.36% |
| NO | 36 | 35.64% | 100.00% |
| Total | 101 | 100.00% | 100.00% |

HVB: hepatitics virus B; HVC: hepatitics virus C; NO: no viral infection.

The results show that the QoL in patients with chronic viral infections was worse than in patients without, as Table 3 shows us.

The presence of cardiovascular comorbidities has a negative impact on the QoL of patients, as shown in Table 4 below.

The majority of patients (61%) presented articular bleeds as the onset symptom, followed by hematomas, nose bleeds and blood in urine, with 44 (43%) of the patients presenting with a mild functional deficit, 22% a moderate deficit, 21% a severe deficit and the rest of 13% had no functional deficit, as can be seen below in Table 5.

In Table 6 we can see that there is an equal number of patients with an absent or mild functional deficit who are having on-demand or prophylactic treatment. There are more patients with a moderate deficit who are on prophylactic treatment (14 out of 22) than on-demand (8 out of 22). Most of the patients with a severe functional deficit are having on-demand treatment (14 out of 21), with only 7 patients out of 21 on prophylactic treatment.

**Table 3.** Quality of Life (QoL) of those who reported chronic viral infections.

| Scale Results of Those Who Reported Viral Chronic Infections | | | | | |
|---|---|---|---|---|---|
| | **Infectious Disease** | **Median** | **Standard Deviation** | **Bartlett Test** | **Difference Meaning** |
| VAS | Absent | 71.74 | 16.63 | $\chi^2 = 0.0015$ $p = 0.97$ | *t*-test Student $t = 2.74$ $p = 0.007$ |
| | Present (HVB, HVC) | 62.26 | 16.54 | | |
| EQ-5D-5L | Absent | 0.23 | 0.23 | $\chi^2 = 0.09$ $p = 0.78$ | *t*-test Student $t = -2.35$ $p = 0.02$ |
| | Present (HVB, HVC) | 0.35 | 0.26 | | |
| Haem-A-QoL-SSS (standardized Scale Score) | Absent | 23.00 | 6.53 | $\chi^2 = 0.002$ $p = 0.97$ | *t*-test Student $t = -1.71$ $p = 0.09$ |
| | Present (HVB, HVC) | 25.31 | 6.48 | | |
| Haemo-SYM | Absent | 34.83 | 17.65 | $\chi^2 = 0.12$ $p = 0.73$ | *t*-test Student $t = -2.09$ $p = 0.04$ |
| | Present (HVB, HVC) | 42.80 | 18.60 | | |
| | Present | 46.97 | 19.20 | | |

VAS: self-evaluation score from EQ-5D-5L.

**Table 4.** QoL of those who reported cardiovascular comorbidities.

| | **Cardiovascular Comorbidities** | **Median** | **Standard Deviation** | **Bartlett Test** | **Difference of Meaning** |
|---|---|---|---|---|---|
| Self-Evaluation | Absent | 68.17 | 16.37 | $\chi^2 = 0.16$ $p = 0.69$ | *t*-test Student $t = 2.49$ $p = 0.01$ |
| | Present | 59.04 | 17.43 | | |
| EQ-5D-5L | Absent | 0.26 | 0.24 | $\chi^2 = 0.43$ $p = 0.51$ | *t*-test Student $t = -3.00$ $p = 0.003$ |
| | Present | 0.42 | 0.21 | | |
| Haem-A-QoL-SSS (Standardized Scale Score) | Absent | 23.60 | 6.05 | $\chi^2 = 1.46$ $p = 0.23$ | *t*-test Student $t = -2.24$ $p = 0.03$ |
| | Present | 26.77 | 7.30 | | |
| Haemo-SYM | Absent | 37.25 | 17.71 | $\chi^2 = 0.26$ $p = 0.61$ | *t*-test Student $t = -2.43$ $p = 0.02$ |

**Table 5.** Functional deficit distribution in our lot.

| **Functional Deficit** | **Frequency** | **Percent** | **Cum. Percent** |
|---|---|---|---|
| Absent | 14 | 13.86% | 13.86% |
| Mild | 44 | 43.56% | 57.43% |
| Moderate | 22 | 21.78% | 79.21% |
| Severe | 21 | 20.79% | 100.00% |
| Total | 101 | 100.00% | 100.00% |

**Table 6.** Functional deficit according to treatment.

| Treatment | Absent | Mild | Moderate | Severe | Total |
|---|---|---|---|---|---|
| On-demand | 7 | 22 | 8 | 14 | 51 |
| Prophylactic treatment | 7 | 22 | 14 | 7 | 50 |
| TOTAL | 14 | 44 | 22 | 21 | 101 |

At diagnosis, 2% of patients were diagnosed with mild hemophilia A, 18% with moderate hemophilia and 80% were severe cases. Even though there is no "evolution" in hemophilia, we tried to assess the efficacy of the prophylactic program by comparing the population by serum levels of factor VIII. Therefore, at the time of the study, we observed that 36% could be categorized as mild cases, 30% as moderate and only 34% as severe hemophilia, if we take into consideration the serum levels of factor VIII. These results, we believe, show that there is a significant improvement in serum levels after the prophylaxis program has been started.

The results from the QoL scales regarding the level of functional deficit, together with the meaning of the differences, are shown below in Table 7.

**Table 7.** Questionnaires scores in regards to functional deficit.

| | Functional Deficit | Median | Standard Deviation | Variance | Bartlett Test | Difference Meaning |
|---|---|---|---|---|---|---|
| VAS | Absent | 83.86 | 9.99 | 99.98 | $\chi^2 = 4.65$ $p = 0.20$ | ANOVA F = 12.45 $p < 0.001$ |
| | Mild | 67.59 | 15.99 | 255.97 | | |
| | Moderate | 60.68 | 12.95 | 167.75 | | |
| | Severe | 54.14 | 16.26 | 264.53 | | |
| EQ-5D-5L | Absent | 0.086 | 0.04 | 0.001 | $\chi^2 = 54.61$ $p < 0.001$ | KRUSKAL–WALLIS $\chi^2 = 36.06$ $p < 0.001$ |
| | Mild | 0.279 | 0.16 | 0.027 | | |
| | Moderate | 0.325 | 0.16 | 0.025 | | |
| | Severe | 0.495 | 0.35 | 0.125 | | |
| Haem-A-QoL-SSS (Standardized Scale Score) | Absent | 17.50 | 3.86 | 14.89 | $\chi^2 = 3.73$ $p = 0.29$ | ANOVA F = 10.59 $p < 0.001$ |
| | Mild | 25.35 | 6.11 | 37.31 | | |
| | Moderate | 23.56 | 5.65 | 31.95 | | |
| | Severe | 28.43 | 6.21 | 38.56 | | |
| Haemo-SYM | Absent | 21.57 | 12.54 | 157.34 | $\chi^2 = 2.52$ $p = 0.47$ | ANOVA F = 12.20 $p < 0.001$ |
| | Mild | 38.41 | 17.74 | 314.85 | | |
| | Moderate | 41.00 | 15.65 | 244.86 | | |
| | Severe | 54.76 | 14.75 | 217.59 | | |

The differences between the results have a high statistical significance; therefore, we can assume that the level of functional deficit is directly linked with QoL. We want to underline a particularity regarding the results obtained with the Haem-QoL scale, where for the mild and moderate levels of functional deficit, we have inversed scores. This may be due to zero being listed for some of the items on the scale.

When applying the EQ-5D-5L questionnaire, and looking at the self-evaluation scale (VAS), we can observe a mean value of 65, which is a good quality of life. In regard to mobility, self-care and activity, the mean value showed moderate trouble, whereas pain and anxiety levels demonstrated mild problems.

With the Haem-A-QoL questionnaire, we utilized the standardized scale score (SSS) and the mean value was 412, as shown in Table 8. This value represents a good quality of life. The highest scores (worst QoL) can be found in the physical, view of oneself and feelings domain. The lowest scores (best QoL) were obtained in the dealing with feelings,

family planning and sexuality domain. The Haemo-SYM had a mean value of 40 points, which represents a good quality of life, as seen in Table 9.

**Table 8.** Evaluation of total standardized score from Haem-A-QoL questionnaire.

| Obs | Total | Mean | Variance | Std Dev |
|---|---|---|---|---|
| 101.0000 | 41,697.4583 | 412.8461 | 23,988.6091 | 154.8826 |
| Minimum | 25% | Median | 75% | Maximum | Mode |
| 91.2500 | 296.4583 | 422.0833 | 520.8333 | 813.5000 | 232.0833 |

**Table 9.** Haemo-Sym results.

| Obs | Total | Mean | Variance | Std Dev |
|---|---|---|---|---|
| 101.0000 | 4044.0000 | 40.0396 | 345.4184 | 18.5854 |
| Minimum | 25% | Median | 75% | Maximum | Mode |
| 4.0000 | 22.0000 | 43.0000 | 53.0000 | 90.0000 | 22.0000 |

When comparing the results obtained by applying the Haem-A-QoL in our study with the studies conducted in Brazil, the total score is worst in our study, showing that the Romanian patients had a lower QoL than the Brazilian patients, but that the most impacted domain (Physical activity) was the same for both countries [10,11].

When performing a linear regression between VAS and the total of EQ-5D-5L the correlation coefficient was 0.59, 0.53 between VAS and Haemo-SYM, and 0.3 between VAS and Haem-A-QoL, which indicates a very good correlation between VAS and EQ-5D-5L and Haemo-SYM.

## 4. Discussion

In regards to the efficacy of treatment programs in Romania [12], we do find an important decrease in severe factor VIII deficiency from 80% to 34%, which is a really good response. This information is backed by the data obtained from the QoL questionnaires, which reveal a good quality of life. The 65 mean value of the VAS scale, represents 65 points on a scale from 0 to 100, with 100 being the best health status possible, and where 65 points on this scale is considered a good QoL. Since the correlation coefficient is quite high between VAL and the three questionnaires, we can state that our patient lot has a relatively good quality of life overall. The most problematic domain in our patient lot was mobility, even if most did not have severe functional problems. Even though the results are only good, they are in the upper part of the scale. When comparing these with results from other countries, we can see that the quality of life in Romania is lower, and therefore improvement is needed. The most affected domain in all studies was the physical domain [13,14].

All three questionnaires are good instruments for measuring quality of life, so at this point, it is up to the physician to decide which they should use.

We believe that the development of comprehensive treatment centers for hemophilia and educational programs can greatly improve disease understanding, patient access to treatment and, therefore, treatment adherence. Moreover, further studies should be conducted after a greater period of time has passed since the prophylactic programs have started to assess if time passed can influence the results.

The preponderance of severe hemophilia is consistent with the findings from other countries and, therefore, allows a good comparison between studies in this regard. The functional deficits of the patients were remarkably good, and better than expected, but those who did have a moderate and severe deficit did not have orthopedic interventions. This may be due to the lack of comprehensive treatment centers, and the lack of orthopedic surgeons qualified for these types of patients in the majority of medical centers, or it could have been because the patients did not require the procedure [15].

Screening for viral infections was introduced in 1992. Prior to this, most of our patients underwent blood transfusions, of which some were positive for viral infections. Therefore, the high percentage, 60%, of patients with HVC infection does not surprise us. Moreover, the results are similar to HVC infections from other countries [16,17]. The aim of this study was not to assess the impact of viral infection on QoL. Since, in international literature, we found an interest in the prevalence of viral infections in hemophilia patients, we thought it would be important to include the prevalence of viral infections in our lot, for a better description. We do expect a decrease in viral infections in the future, as new viral infections due to transfusion are not expected [18].

## 5. Limitations of Study

We do want to address the issue that the patients included in this study were those who have access to the two excellent treatment centers for hemophilia in the western part of the country. Since hemophilia care is not balanced throughout the country, a larger lot, with patients from different counties, would be advised.

## 6. Conclusions

The QoL of hemophilia patients in Romania was found to be relatively good at the time of this study; however, the collection of data from a larger lot of patients is advised.

**Author Contributions:** Conceptualization, H.M.I.; Methodology, I.I.; Software, validation and formal analysis: M.-G.H.; Investigation, C.G., A.-M.I. and A.T.; Resources, C.G.; Writing, C.G.; Review and editing, H.M.I., M.-G.H. and I.I. All authors have read and agreed to the published version of the manuscript.

**Funding:** This research received no external funding.

**Institutional Review Board Statement:** The study was conducted in accordance with the Declaration of Helsinki, and approved by the Ethics Committee of the University of Medicine and Pharmacy "Victor Babes" Timisoara.

**Informed Consent Statement:** Informed consent was obtained from all subjects involved in the study.

**Data Availability Statement:** Not applicable.

**Acknowledgments:** We would like to thank Anne Rentz, EuroQoL Foundation, and the Haem-QoL Project for the permission to use the questionnaires used in this paper, and the tools needed to process the information. Most of all, we would like to thank the patients for their availability to take part in this study.

**Conflicts of Interest:** The authors did not receive funding for this research, and are not employed by any pharmaceutical company, but have received speaker fees from companies such as: Janssen & Janssen, ABBVIE, Takeda, Novartis, Sanofi-Genzyme, Roche, Novo Nordisk, Accord, etc.

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
