# Peer review of "A Perspective on the Quality of Life of Hemophilia A Patients in Romania—A Study on 100 Patients"

_2813-2475, doi:10.3390/jvd1020012_

Round 1

Reviewer 1 Report

Thank you for allowing me to review your article entitled "A perspective on the quality of life of hemophilia A patients in Romania - a study on 100 patients".  Quality of life is a very important topic in regard to the care of persons with hemophilia so understanding the issues the affect it are key.

After reviewing your manuscript, I do feel that are some edits that would enhance the manuscript prior to publication.

Introduction:

- Please change hemophilia A patients to "patients with hemophilia A".  patient first language is preferred.  Please ensure this is corrected through out the manuscript as well.

Abstract:

Introduction:

Overall I would condense the introduction and change the focus. The first 2 and 4th sentence are not really needed and can be deleted and then change the rest of the focus to what is happening in Romania.  

Would just talk about quality of life in hemophilia and what is known. in general  Then talk about in romania prophylactic therapy was not available until 2017 so QOL was difficult to asses and when it was it showed poor quality.  Then talk about why your paper is important and what you are doing.  something like now that prophylaxis has been available since 2017, we aimed to evaluate the quality of life in persons with hemophilia in romania since it was instituted.

Other Abstract Comments:

Sentence 1: technically Fresh frozen plasma has been available to treat Hemophilia since the 1940s so the statement about treatment began in 1971 is inaccurate.  Need to modify this sentence

Sentence 2 and 3- please change "recombined" to "recombinant"

Results:

The majority of the results prose is just descriptions of the patient population.  I believe that the vast majority of the prose could be condensed into a single demographics table.

It is also important to break the data into the whole population but then into mild vs moderate vs severe and on demand vs prophylaxis patients in the severe category so that limitations can be assessed more easily.  Break down the functional deficit data into these categories.

The viral table data can be included in the demographics table.

More data is needed on the QOL questionnaire results.  No inferences can be made based on the way in is presented.  There is not a good description of the scoring and what a mean of 412 really means.  This needs to be elaborated on in the materials and methods sections.  i would make a table of all of the results and then describe the most important ones in the prose.  As it is written it really isnt clear what the QOL was reported as in patients with hemophilia in Romania.  

Discussion:

The first paragraph discusses life expectancy.  This concern may change QOL but it wasnt discussed in the Results section so it seems out of place.  If this was a reported concern in the questionnaires then discuss here, but if it wasnt then there is a lot of other data to discuss/clarify already.

In the second paragraph, you state that functional deficits were actually good/better than expected, but then you lament the lack of orthopedic interventions.  It is important to understand the lack of surgical providers, but maybe the lack of interventions was because functionally the risk out weighed the benefit for the patient as well.  They may not need surgical intervention if they do not feel they have functional deficits.

Third paragraph discusses viral diseases, but not in terms of QOL.  Did patients with known viral infections have worsened QOL.  If that was shown, then discuss that her and why that might be the case.  Just discussing the viral infections in the population does not help explain QOL issues/concerns.

THe 4th paragraph should be the first in the discussion.  Discuss what the QOL surveys showed, then how Romania compares then use the rest of the discussion to discuss what aspects of life in Romania for patients with hemophilia affect it the most and postulate on how it might be improved/fixed.

Author Response

Thank you for taking from your valuable time to help us with our paper.

Introduction: We have made the alterations suggested by you. If you find them insufficient please let me know and I will try to improve them further.

Sentence 1: technically Fresh frozen plasma has been available to treat Hemophilia since the 1940s so the statement about treatment began in 1971 is inaccurate.  Need to modify this sentence- this was the data I found. Thank you for pointing it out.

The majority of the results prose is just descriptions of the patient population.  I believe that the vast majority of the prose could be condensed into a single demographics table.- table done and added. If still to long I will shorten it further. I have a tendency to over-do...

Break down the functional deficit data into these categories.= done

The viral table data can be included in the demographics table - This I didn't modify, as the other reviewer thought the paper should contain a bit more details about comorbidities.

More data is needed on the QOL questionnaire results.  No inferences can be made based on the way in is presented.  There is not a good description of the scoring and what a mean of 412 really means.  This needs to be elaborated on in the materials and methods sections.  i would make a table of all of the results and then describe the most important ones in the prose.  As it is written it really isnt clear what the QOL was reported as in patients with hemophilia in Romania. - I hope the modifications done will help with the issue stated by you.

Discussion: all the alterations suggested by you were done.

Thank you kindly for your time and guidance! I hope you will have a wonderfull and peacefull weekend!

Best wishes, Catalina.

Reviewer 2 Report

I have no understanding how the below happened:

"At diagnosis 2% of patients were diagnosed with mild hemophilia A, 18% with moderate 91 hemophilia and 80% were severe cases. At the moment of the study we cathegorised by 92 value of serum levels of factor VIII the evolution of patients, and we observed that 36% 93 could be cathegorised as mild cases, 30% as moderate and only 34% as severe hemophilia. "

There is some misunderstanding. There is no "evolution" in haemophilia A.

The gualuty of life of patients with HBV /HCV is low, irrespective of haemophilia. The prevalence in the study is very high. This should be analyzed and explained.

Can co-morbidities (e.g. cardio-vascular) /other health problesm be analyzd and presented?

The numbers /results from used instrument tell me nothing.

The readred need to understand the score, the differences should be maybe shown graphically with division on areas.

How is mild /moderate etc. deficit defined.

English must be improved.

Author Response

I have no understanding how the below happened:

"At diagnosis 2% of patients were diagnosed with mild hemophilia A, 18% with moderate 91 hemophilia and 80% were severe cases. At the moment of the study we cathegorised by 92 value of serum levels of factor VIII the evolution of patients, and we observed that 36% 93 could be cathegorised as mild cases, 30% as moderate and only 34% as severe hemophilia. "

There is some misunderstanding. There is no "evolution" in haemophilia A. Indeed there is no "evolution" but we tried evaluating the efficacy of prophilactic treatment by following the serum levels of factor VIII. We made alterations to this paragraph. I hope that our target is better understood now. 

The gualuty of life of patients with HBV /HCV is low, irrespective of haemophilia. The prevalence in the study is very high. This should be analyzed and explained.

Also, this was corrected. I hope to your liking.

Can co-morbidities (e.g. cardio-vascular) /other health problesm be analyzd and presented?- we added

The numbers /results from used instrument tell me nothing.

The readred need to understand the score, the differences should be maybe shown graphically with division on areas.- we made alterations to the material and methods. I believe it should be more appropriate.

How is mild /moderate etc. deficit defined.- also added in the material and methos sections

Round 2

Reviewer 2 Report

The overall interest is moderate but the presentation is improved now.